# The Fundamental Role of Lipids in Polymeric Nanoparticles: Dermal Delivery and Anti-Inflammatory Activity of Cannabidiol

**DOI:** 10.3390/molecules28041774

**Published:** 2023-02-13

**Authors:** Mark Zamansky, Na’ama Zehavi, Amnon C. Sintov, Shimon Ben-Shabat

**Affiliations:** 1Department of Biomedical Engineering, Ben Gurion University of the Negev, Be’er Sheva 84105, Israel; 2Laboratory for Biopharmaceutics, E.D. Bergmann Campus, Ben-Gurion University of the Negev, Be’er Sheva 84105, Israel; 3Department of Biochemistry and Pharmacology, Ben Gurion University of the Negev, Be’er Sheva 84105, Israel

**Keywords:** cannabidiol, lipid-stabilized nanoparticles, ethyl cellulose, skin permeability, interleukin release, HaCaT cell culture

## Abstract

This report presents a nanoparticulate platform for cannabidiol (CBD) for topical treatment of inflammatory conditions. We have previously shown that stabilizing lipids improve the encapsulation of CBD in ethyl cellulose nanoparticles. In this study, we examined CBD release, skin permeation, and the capability of lipid-stabilized nanoparticles (LSNs) to suppress the release of IL-6 and IL-8. The nanoparticles were stabilized with cetyl alcohol (CA), stearic acid (SA), lauric acid (LA), and an SA/LA eutectic combination (SALA). LSN size and concentration were measured and characterized by differential scanning calorimetry (DSC), in vitro release of loaded CBD, and skin permeability. IL-6 and IL-8 secretions from TNF-α-induced HaCaT cells were monitored following different LSN treatments. CBD released from the LSNs in dispersion at increasing concentrations of polysorbate 80 showed non-linear solubilization, which was explained by recurrent precipitation. A significant high release of CBD in a cell culture medium was shown from SALA-stabilized nanoparticles. Skin permeation was >30% lower from SA-stabilized nanoparticles compared to the other LSNs. Investigation of the CBD-loaded LSNs’ effect on the release of IL-6 and IL-8 from TNF-α-induced HaCaT cells showed that nanoparticles stabilized with CA, LA, or SALA were similarly effective in suppressing cytokine release. The applicability of the CBD-loaded LSNs to treat topical inflammatory conditions has been supported by their dermal permeation and release inhibition of pro-inflammatory cytokines.

## 1. Introduction

It has previously been shown that cannabidiol (CBD) may be effective in treating dermal diseases such as psoriasis, with an advantage of having a different mode of therapeutic action [1,2]. CBD possesses an anti-inflammatory activity while corticosteroids and vitamin D_3_ analogs act through an antiproliferative mechanism. CBD was shown to inhibit TNF-α-induced NF-kB transcription in a dose-dependent manner in HaCaT cells [3]. It has therefore been conceived that combining the advantages of CBD as an anti-inflammatory agent with an appropriate dermal delivery system with enhancing skin permeation would implement its therapeutic potential. This may be so since novel approaches in the design of topical medications are the key to successful treatment of chronic skin diseases with unmet needs. Although the pharmacological mechanisms are well known in many dermatological diseases, treatments that are more adequate have still been required. In general, the provided treatments should involve enhanced penetration and delivery of the active material into the skin and prolonged retention at the site of action. Dermal and transdermal delivery technologies include electroporation and micro-electroporation [4], thermal ablation [5], sonophoresis and iontophoresis [6,7], the use of chemical enhancers [8], and nanoparticles [9]. The latter method is particularly interesting since the incorporation of CBD into nano-carrier systems improved its stability, permeability, and bioavailability [10,11]. Besides their capability to penetrate the stratum corneum, nanoparticles can also provide a prolonged release profile of the active agent along with the traditional skin application (such as a gel product). A possible mechanism for the penetration and retention of nanoparticles (NPs) in the skin includes an entry of the applied nanoparticles into the hair follicles [12,13,14] and a permeation between and through the corneocytes [15,16].The composition of the NPs usually affects the extent of permeation and the rate of drug release; thus, skillful compounding of the nanoparticles enables adjustment and fine-tuning of permeation and release properties. For example, gold nanoparticles were shown to permeate through pig skin due to their nanoscale size [17], whereas polymeric nanoparticles such as PLGA-NPs additionally enhanced the permeation of flufenamic acid through the skin by a change in the local pH that mediates the hydrolytic degradation rate of the polymer [18]. Another report showed that the release profiles of solid lipid nanoparticles (SLNs) prepared with a combination of theobroma oil and beeswax were depended on the melting point of the components [19]. In a release study performed at 37 °C, initial quick release rates of loaded drugs were associated with theobroma oil having a melting point between 34–38 °C, whereas further slower release rates were associated with beeswax’s melting point of 63–67 °C. Another example of the influence of lipids and their combinations on drug release profiles was shown in ketoprofen-loaded SLNs prepared with beeswax and carnauba wax. Higher carnauba-wax content in the nanoparticles resulted in slower ketoprofen release rates [20]. In another study, stearyl amine and Esterquat 1 provided different release profiles for the loaded active agent [21]. It was also shown that the release of cannabidiol (CBD) from PLGA-NPs was dependent on the production mode, i.e., NPs prepared by a direct co-precipitation technique gave a burst release, whereas NPs prepared by the emulsification-evaporation method showed a more moderate rate of CBD release [22].

Skin retention of an active agent, rather than rapid permeability through the skin, would be beneficial for local pathological conditions involving bacterial or dermatophyte infection (e.g., acne) and autoimmune conditions (e.g., psoriasis). In other words, there is a need for a topical treatment of psoriasis or acne, wherein the drug should minimally be exposed to systemic circulation. The current topical treatment of psoriasis is based on potent corticosteroids, vitamin D_3_ analogs, vitamin A derivatives, and their combinations. Although a safety concern exists for the long-term usage of these treatments, they have still been frequently applied for prolonged periods due to the absence of alternatives. Vitamin D_3_ analogs, especially calcipotriene, may cause itching, local irritation, and sometimes worsening of the psoriatic condition, while corticosteroids may cause epidermal thinning, striae, and dermal infections, and are generally not recommended for prolonged application [23,24]. Thus, it is essential to seek better alternatives to the current topical treatments that reduce local adverse effects and improve patient compliance [25].

In our previous work, we have successfully formulated CBD-loaded polymeric nanoparticles, based on ethylcellulose and a stabilizing lipid, with a controllable narrow range of particle size and a high encapsulation efficiency [26]. These CBD-loaded nanoparticles efficiently delivered CBD into the cytosol of HaCaT cells. In the present paper, we report a relationship existing between the stabilizing lipids incorporated in the nanoparticles and the skin penetrability of CBD. We also present the effect of CBD-loaded nanoparticles prepared with different stabilizing lipids on cell viability and their ability to reduce TNF-α-induced inflammation in HaCaT cells.

## 2. Results and Discussion

### 2.1. Nanoparticle Preparation Method and Characterization

The current research aimed to prepare and optimize CBD-loaded lipid-stabilized nanoparticles (LSNs) for dermal delivery and treatment of dermatological conditions such as psoriasis. In addition, we studied the permeability and effectiveness of different CBD-loaded LSNs and their lipid combinations. Cetyl alcohol (CA) which has been found previously as an efficient stabilizer of CBD-loaded NPs, was tested and compared with two other lipids, stearic acid and lauric acid (SA and LA) [26]. These lipids are suitable for nanoprecipitation since they are soluble in the polymer ethanolic solution and possess dermal permeation enhancement properties [27,28,29]. In addition, a combination of stearic and lauric acids (SALA) was tested at a eutectic ratio of 24:76 [30]. The use of eutectic mixtures is based on previous examples, such as the enhanced dermal permeability of lidocaine and prilocaine combination [31], and excipients of Deep Eutectic Solvents (DES) [32,33]. Permeability enhancement of eutectics is usually explained by melting-point depression and was supported by mathematical models [34].

The four CBD-loaded and unloaded LSN types (NP-CA, NP-SA, NP-LA, and NP-SALA) were prepared by the non-solvent (water) precipitation method as previously described. As seen in Table 1, no profound differences in size were obtained between the CBD-loaded and unloaded NPs stabilized with the same lipid, which indicates that the creation of the nanoparticle network is not affected by small molecules. However, it seems that the nature of the stabilizing lipid directly influences the obtained NP size, particularly the less water-soluble fatty alcohol CA and more soluble fatty acids LA, SA, and their combination SALA. In our previous work, we have shown that inclusion of materials with lower water solubility contributes to NP size reduction by creating a larger number of nucleation centers [26]. Thus, the polymer is distributed between a large number of particles at a smaller size. NPs stabilized with CA were the smallest in size, whereas those stabilized with SA were the largest. A statistically significant difference (*p* < 0.05, ANOVA test) was observed between these two NP types. All the nanoparticles were prepared with the same concentrations of polymer, stabilizing lipids, and CBD, i.e., 0.05% *w*/*w*, 0.025% *w*/*w*, and 0.015% *w*/*w*, respectively, except for the LA-stabilized NPs, for which 0.05% *w*/*w* LA was used, due to insufficient stabilization at its lower concentration.

The degree of crystallinity of CBD in the LSNs, ranging from amorphous to a crystalline form, is one of the parameters that affect CBD release and dissolution kinetics. In a previous paper, it was reported that the absence of crystalline forms allows quick release and permeation of the loaded drugs [35]. The thermograms in Figure 1 represent DSC scans of stabilizing lipids, ethyl cellulose, CBD, and the related LSN preparations. The melting or transition points recorded as peak maxima corresponded to the known values for all components except for ethyl cellulose (see Table 2), which deviated by about 30 °C. This decrease in the glass transition temperature of ethyl cellulose may be due to the high moisture content of the analyzed sample. The CBD-loaded NPs stabilized with all four types of lipids did not exhibit any endothermic peak between 30 °C and 169 °C (169 °C relates to mannitol’s melting point (m.p.) used in the formulation as a lyophilization cryoprotectant), indicating a complete absence of crystalline structures of either CBD or lipids. An absence of ethyl cellulose endotherm peak suggests that the polymer itself is also amorphous and lacks crystalline regions. Considering these results, we can assume that the nanoparticle structure is a homogeneous mixture of components rather than a separate CBD–lipid core and a polymeric shell or vice versa.

### 2.2. CBD Release from LSNs

Due to the poor water solubility of CBD (about 0.1 µg/mL), the release at a “sink” condition required the addition of a surfactant to the release medium for its solubilization [41]. Following other reports, which have evaluated the release of CBD from nano- and microparticles, we used polysorbate 80 [22,42]. Polysorbate 80 is one of the frequently used surfactants in the pharmaceutical industry to assess the release and dissolution of poorly soluble drugs. According to a surfactant-assisted solubility model [43], the general equation for micellar solubilization is:(1)$$S_{tot}=S_{w}+k\left(C_{surf}-CMC\right)=S_{w}+kC_{mic}$$
where $S_{tot}$ is the total solute solubility, $S_{w}$ is the water solubility, k is the surfactant solubilization capacity, $C_{surf}$ is the surfactant concentration, CMC is the critical micellar concentration, and $C_{mic}$ is the micellar surfactant concentration. If the surfactant concentration is much greater than the CMC, which is valid for most surfactants in the pharmaceutical industry, the above equation can be transformed to
(2)$$S_{tot}\approx \;S_{w}+kC_{surf}$$

Thus, it is reasonable to expect a linear relationship between the surfactant concentration and the total solubility. A dissolution experiment with an excess CBD (powder) in PBS at 37 °C in the presence of 0.1% *w*/*w* polysorbate 80 revealed that 70–75 µg of CBD were dissolved per one milliliter. Then, we performed preliminary release experiments with CBD-loaded EC nanoparticles stabilized with CA. The experiments were carried out at six concentrations of polysorbate 80: 0.1%, 0.5%, 1.0%, 3.0%, 5.0%, and 10.0% (*w*/*w*). The CBD concentration was 5 µg/mL in applied LSNs (7.5 × 10^9^ nanoparticles/mL), which is more than ten times lower than CBD solubility (70–75 µg/mL in PBS + 0.1% PS-80, 37 °C). The release medium, therefore, reasonably represented a perfect “sink”. As shown in Figure 2A, the release patterns of CBD were similar at the various concentrations of polysorbate 80, demonstrating a quick release in the first 15 min and a plateau until 24 h. The 24 h release of the CBD content in the LSNs seemed to be incomplete, as the maximal concentrations of CBD in the release media were 1.2 µg/mL (20% of total CBD), 2.2 µg/mL (50% of total CBD), and 2.8 µg/mL (60% of total CBD) at 0.1%, 0.5%, and 1.0% polysorbate 80, respectively. However, by increasing the surfactant concentration to 3.0%, 5.0%, or 10% (*w*/*w*), the total release of CBD increased to approximately 80%, independent of the surfactant level. This finding, which does not follow Equation (2), is also in disagreement with several previous publications that reported a gradual and complete release of CBD over several hours using ≤0.5% *w*/*w* concentrations of polysorbate 80 [22,42].

To explain this discrepancy, it is postulated that the non-degradable EC nanoparticles may form nucleation seeds that facilitate CBD precipitation. It is a reasonable explanation since previous studies have shown that cellulose polymers could serve as precipitation inhibitors and vice versa, nucleation and crystallization accelerators [44,45]. Unlike non-degradable cellulose nanoparticles, complete and surfactant-dependent release and solubilization of CBD can be achieved when degradable polymers such as PLGA nanoparticles [22] and polycaprolactone microspheres [42] are studied in surfactant-containing release media. DLS size measurements confirmed the precipitation of CBD structures during the release process. An approximately ×2 increase in particle size and an increase in the size distribution range of nanoparticles were noted after 24 h incubation in PBS/polysorbate 80 medium. This increase implies that the CBD re-precipitation was also accompanied by aggregation of the nanoparticles.

According to the solubilization mechanism proposed by Shaeiwitz et al. [46], the micelle adsorbs onto the surface of the particle with the re-precipitated CBD, forming a complex with one or several CBD molecules, and desorbs. Considering an equilibrium state, CBD deposition from the CBD-micelle complex would work similarly. Since the observed micellar solubility in the presence of particles is lower than for a particle-free system, it could be assumed that the deposition rate for a single CBD molecule is significantly higher than the desorption rate leading to a low micelle-solubilized CBD concentration. However, the assumption that each CBD molecule can be solubilized or deposited by a micelle, independently of other CBD molecules occupying this micelle, should also lead to a linear relationship between micelle concentration and solubilized CBD. The “saturation” phenomenon presented in Figure 2A could be explained by the mutual stabilization of CBD molecules occupying the same micelle. Such stabilization would reduce the desorption rate from precipitated CBD structures, requiring several successful consecutive desorption events.

Assuming that *a* moles of CBD are successfully desorbed (a>1) for each mole of micelles, the following process is proposed: $aCBD_{s}+Mic\;\overset{K}{↔}\;aCBD_{mic}$, where $CBD_{s}$ is the precipitated CBD structures, Mic is the micellar surfactants, $CBD_{mic}$ is the micelle-solubilized CBD, and *K* is the equilibrium constant. Therefore, the equilibrium relationship between solubilized CBD and micelle concentration would be:$$K=\frac{{\left[CBD_{mic}\right]}^{a}}{{\left[CBD_{s}\right]}^{a}\cdot \left[Mic\right]}$$
$$\left[Mic\right]=C_{mic}$$
(3)$$\left[CBD_{mic}\right]=\left[CBD_{s}\right]\cdot \sqrt[{a}]{\left(K\cdot C_{mic}\right)\;}$$

If $f_{s}$ is the fraction of the precipitated CBD, then:(4)$$f_{s}=\left[CBD_{s}\right]/\left(\left[CBD_{s}\right]+\left[CBD_{mic}\right]\right)$$

Considering that $C_{mic}=\left(\left[PS-80\right]-PS-80_{CMC}\right)/N_{ag}$, where [PS−80] is the concentration of polysorbate 80, $PS-80_{CMC}$ is the CMC value of polysorbate 80, $N_{ag}$ is its aggregation number (experimentally reported values ranging between 22–350 [47]), and $\left[PS-80\right]\gg PS-80_{CMC}$ ($PS-80_{CMC}$ value is about 0.0014% *w*/*w* [48]), then Equations (3) and (4) can be combined:(5)$$\sqrt[{a}]{K\cdot \left[PS-80\right]/N_{ag}}+1=1/f_{s}$$

The relationship derived from Equation (5) is shown in Figure 2B, using a=2, and cumulative CBD release data obtained after 24 h.

Once the initial conditions had been set up, release experiments of CBD from the LSNs were conducted in PBS and DMEM, a cell culture medium used in the bioassay discussed in Section 3.7. An appropriate quantity of LSNs was added to the release medium (containing 0.05% polysorbate 80) to achieve a total CBD concentration of 1 μg/mL (instead of 5 μg/mL applied in prior experiments). The sampling intervals were shortened to allow a better resolution of the initial release phase.

As presented in Figure 3A, it was found that incorporation of a relatively higher proportion of ethyl cellulose (EC/CBD ratio of 20:1) expectedly reduced the release rate of CBD compared to LSNs prepared with a lower concentration of EC (EC/CBD ratio 10:3). A higher EC content also decreased the total cumulative release of CBD at 2 h (Figure 3A, CA-NP EC/CBD 20:1 vs. CA-NP EC/CBD 10:3). The Higuchi release model, which depends on extending depleting zones from surface to core, may describe this finding. According to the Higuchi model, $f_{rel}=K_{H}\sqrt{t}$, where $f_{rel}$ is the released fraction, and $K_{H}$ is the Higuchi constant [49]. Figure 3B presents this correlation demonstrating the suitability of the release data to the model for CA- and SA-stabilized LSNs. The release pattern that was fitted to the Higuchi model implies that the NPs are formed as a polymeric matrix containing an evenly distributed active agent. This contrasts with a CBD core entrapped by a polymeric shell, in which the release should be linear as long as C*_initial_* >> C_s_. The slower release as a function of polymer content has also been demonstrated in several publications describing release from EC microparticles and nanoparticles [50,51,52]. It can be seen that CBD release from the NPs with the higher content of polymer (EC:CBD 20:1, blue triangles) fits relatively better to the Higuchi model, probably due to their larger size and more likely due to the uniform distribution of CBD as also demonstrated by the DSC measurements (Figure 1).

These studies have shown that similar to large oral dosage forms, the increase in EC content reduces the release rate of the entrapped drugs. The reduced level of the plateau segment of the release pattern for nanoparticles with an EC:CBD ratio of 20:1, may be related to an increase in the CBD precipitation rate due to the relatively larger surface area of these particles. As shown in Figure 3A, CA- and SA-stabilized NPs have similar release profiles, with a fast release in the first minutes and a slight precipitation at 2 h. The LA-containing LSNs also released CBD at an initial fast rate, but the second phase of the cumulative solubilized CBD concentrations during a 2 h period was slightly lower than that obtained with CA- and SA-stabilized LSNs. However, the three types of LSNs (stabilized by CA, SA, and LA) reached maximum release levels ranging between 82% to 84%. Unlike these LSNs, almost no CBD precipitation was noted at 2 h when SALA-containing NPs were tested, as 97% of CBD content was solubilized.

To evaluate the release pattern of the LSNs in a cell culture medium, CBD release experiments were performed in DMEM for CA- and SALA-stabilized nanoparticles (at a total of 1 µg/mL CBD concentration). Since the solubility of CBD in DMEM is approximately 1 µg/mL, the release was performed under a “non-sink” condition [26]. As seen in Figure 4, slower release rates and an incomplete release were observed for both types of LSN. However, the precipitation rate of CBD released from SALA-stabilized NPs was relatively slower, and after 2 h, the released CBD content was more than twice as high as that released from CA-stabilized NPs. The decrease in the solubilized CBD in DMEM can be explained by an increased precipitation rate induced by a high content of protein and colloidal particles combined with the low solubility of CBD in a fully supplemented DMEM. The difference in precipitation rates between CA- and SALA-stabilized NPs may be attributed to the difference in melting points of CA and SALA, 51 °C and 36 °C, respectively. Since the SALA melting point was below the experimental temperature, it can be assumed that SALA was released into the system in its liquid form, facilitating the solubilization of CBD.

### 2.3. Dermal Retention and Transdermal Permeability of Lipid-Stabilized Nanoparticles

The permeability of CBD in the lipid-stabilized NPs into and through skin was tested in vitro using excised rat skin. Each LSN sample was diluted with deionized water to obtain a concentration of 500 µg CBD/mL, and 0.2 mL of this dispersion was applied onto the skin, providing a 100 µg dose of CBD per 1.77 cm^2^ skin surface area (100 µg CBD/0.2 mL volume or 56.5 µg/cm^2^ skin surface area). Usually, topical formulations are applied at 4–25 mg/cm^2^, thus unlike a gel or ointment, which can be massaged on the skin to increase penetration, the unstirred volume of these nano-dispersions (0.2 mL) was relatively excessive for normal dermal application as only a small portion of LSNs came in direct contact with the skin [53]. Despite that, a quantitative penetration of CBD into the inner layers of fresh rat skin was detected, which was also related to the type of stabilizing lipid. Figure 5 presents the extent of CBD accumulation in rat skin at 6 h of LSN applications (Figure 5A), as well as the kinetics of permeation through the skin into the receiver (Figure 5B). As shown in Figure 5A,B, SA-stabilized NPs demonstrated a relatively low penetrability of CBD into the skin. While the intradermal levels of CBD loaded in CA-NPs, LA-NPs, and SALA-NPs after 6 h were similar, the SA-NPs showed more than 30% lower CBD retention in the skin, compared to NPs stabilized with other lipids. Compared to SA-stabilized NPs, CBD penetration by CA-stabilized NPs showed a significant difference (*p*-value < 0.01, *t*-test). Compared to LA-stabilized and SALA-stabilized NPs, CBD penetration by SA-stabilized NPs showed no statistically significant difference (*p* = 0.055, *t*-test). Interestingly, CBD-loaded CA-stabilized NPs provided significantly higher cutaneous penetration and also percutaneous permeation compared with those obtained by CBD-loaded SA-stabilized NPs (*p* < 0.05, *t*-test).

These results may be explained by using the thermodynamic solubility and activity theory of permeation [54]. According to this theory, the permeation extent depends on the activity of the drug or the active agent in its formulation, whereas activity depends on its saturation ratio, defined as the ratio between actual and saturation solubilities. An active agent with a saturation ratio close to one or above it (supersaturation), would have a higher activity and, therefore, would permeate better. Thus, in a case where the active agent is less soluble in the vehicle and the formulation, its skin permeation would be higher. For example, Casiraghi et al. [30] have shown that the permeation of CBD in two types of formulations, lipophilic and hydrophilic, can be explained by using the Hildebrand solubility parameter δ for the separate formulation components, the whole formulation, and skin ceramides [30]. A higher intradermal and transdermal permeation of CBD was observed with vehicles in which the CBD solubility is relatively low, and the difference in theoretical solubility distance $\Delta \delta^{2}$ is relatively larger. Since the stabilizing lipids (CA, SA, LA, and SALA) in LSNs are all solid at room temperature, and therefore a direct measurement of CBD solubility is not possible, we have used a solubility parameter instead. We preferred to use Hansen solubility parameters (HSP), which had been derived from Hildebrand solubility parameters [55]. HSP has a better ability to explain solubility phenomena by taking into account the hydrogen-bonding interaction term. The model uses the HSP to characterize each component of the formulation as well as the skin.

Table 3 provides the individual HSP values and the calculated distance. The value taken for skin was calculated for human skin and used here as the best available substitute for the rat skin value as was used in the experiment [56].

The summary of calculated values in Table 3 shows that CA and its formulation have the lowest affinity to CBD, whereas LA has the highest. Therefore, CBD-containing NPs stabilized with CA should have the highest activity in terms of CBD saturation, and consequently, the highest skin permeability was obtained. Nevertheless, this model does not explain the low skin penetrability of CBD delivered by SA-stabilized NPs. The relatively high melting point of SA may explain this discrepancy. Remaining in a solid state, SA may hinder CBD penetration into the skin. Similarly, the lower melting points of LA and SALA contribute to the release of CBD, whereas a higher affinity decreases CBD activity in the formulation. A combination of these two factors with an opposite influence can explain higher variability in the permeation of CBD in NPs stabilized with either LA or SALA. Interestingly, the HSP distance between CBD and skin is larger than the distances of each of its mixtures with the tested lipids to skin. Therefore, a higher affinity and a better partitioning between the skin and the aqueous dispersion system are obtained. Compared with pure polymeric NPs, therefore, the incorporation of stabilized lipids into NPs can obviously enhance the skin penetration of active molecules. The same combined consideration of affinity and melting point also explains the lesser extent of re-precipitation of CBD from SALA-stabilized nanoparticles as discussed in the previous section.

### 2.4. Cell Assays

#### 2.4.1. Viability Testing

An XTT viability assay was used to determine the concentrations at which CA-, LA- and SALA-stabilized nanoparticles (CBD-loaded and unloaded LSNs) were toxic to HaCaT keratinocytes. It has been previously shown that free CBD was not toxic to HaCaT cells at concentrations up to 20 µM after incubation for 24 h and 48 h [58]. Our group has also shown that CBD and NPs loaded with low concentrations of CBD had no influence on cell viability at low concentrations, however, unloaded NPs did reduce the cell viability [26]. In the current study, we applied LSN preparations containing high and low loads of CBD per 10^12^ NPs. CBD loading concentrations were 1797 µg and 542.3 µg per 10^12^ CA-stabilized NPs, 1332.8 µg and 504.1 µg per 10^12^ LA-stabilized NPs, and 617.5 µg and 542.9 µg per 10^12^ SALA-stabilized NPs. The in vitro study was performed at CBD concentrations ranging between 5.7 µM and 690 µM, between 5.3 µM and 640 µM, and between 2.5 µM and 690 µM, for CA-, LA- and SALA-stabilized nanoparticles, respectively. The HaCaT cells were treated with the LSNs for 24 h.

Figure 6A compares the cell viability percentages versus CBD concentrations. It can be seen from this figure that the viability was significantly higher than 20 µM (a limit level for CBD determined by Petrosino et al.), when the cells were treated with CBD loaded in the LSNs [58]. the viability profiles of the HaCaT cells as a function of the applied CBD concentration (in LSNs) (as shown in Figure 6A) fitted approximately to each other for all types of the tested LSNs, suggesting that the drop in viability was only slightly related to the CBD concentration and rather more related to the LSN type and the CBD loading concentration per number of nanoparticles (Figure 6B). Calculated IC-50 values for CA-stabilized NPs with a high and a low CBD concentration were 239.1 ± 32.3 µM and 278.6 ± 44.8 µM, respectively. IC-50 values for LA-stabilized NPs with a high and a low CBD concentration were 153.6.1 ± 32.1 µM and 130.7 ± 22.7 µM, respectively. These results indicate that LA-stabilized NPs were significantly more cytotoxic than the CA-stabilized NPs (*p* < 0.05 comparing high loads, and *p* < 0.01 comparing the low loads, ANOVA test). In cells treated with SALA-stabilized NPs, the viability did not drop below 80% up to a CBD level of 690 µM, suggesting that their IC-50 is above this value. By plotting the percentages of cell viability against LSN concentrations (Figure 6B) it is shown that the cell viability depends on the concentration of the CBD loaded per 10^12^ NP. As shown in Figure 6B.1.,6B.2, the highest IC-50 values were obtained by the unloaded CA-stabilized NPs and the LA-stabilized NPs. In addition, the cytotoxicity of the unloaded CA-stabilized NPs was comparable to that obtained by the same LSNs loaded with 542.3 µg CBD per 10^12^ NPs, while both were significantly less cytotoxic than the same LSNs loaded with 1797 µg CBD per 10^12^ NPs. Unlike CA-stabilized NPs, a significant difference was observed between unloaded LA-stabilized NPs and the same LSNs loaded with 504.1 µg CBD per 10^12^ NP and with 1322.8 µg CBD per 10^12^ NPs. There was also a significant difference between these LSNs loaded with 504.1 µg CBD per 10^12^ NP and LSNs loaded with 1322.8 µg CBD per 10^12^ NP (*p* < 0.01, ANOVA test). LA-stabilized NPs, loaded with both high and low CBD, had a significantly lower IC-50 compared to unloaded CA-stabilized NPs or loaded with 542.3 µg CBD per 10^12^ NPs (*p* < 0.05, *t*-test). A comparison of IC-50 values of unloaded CA-LSNs (IC_50_ = 180.9 ± 14.0 × 10^9^ NPs/mL) and unloaded LA-LSNs (IC_50_ = 324.6 ± 52.5 × 10^9^ NPs/mL) showed that LA-LSNs were significantly less cytotoxic than CA-LSNs (*p* < 0.05, *t*-test).

According to Figure 6B.3, there were no significant differences in viability between loaded and unloaded SALA-stabilized NPs. Although IC-50 values could not be calculated for SALA-stabilized NPs at the NP concentration range in this experiment, it is at least higher than 400 × 10^9^ NPs/mL. To summarize this section, the cytotoxicity of CBD-loaded LSNs can be rated as LA-LSNs > CA-LSNs > SALA-LSNs, while the unloaded LSNs are rated as CA-LSNs > LA-LSNs > SALA-LSNs. Based on the viability results, the constant treatment conditions for the next experiments with inflamed HaCaT keratinocytes (Section 3.7.2) were set up using maximal CBD and LSN concentrations that had not affected the viability of these cells, i.e., 105 µM CBD according to CBD-loaded LA-stabilized NPs, and 25 × 10^9^ NPs/mL according to the viability pattern obtained by the CA-stabilized NPs with high CBD loading.

#### 2.4.2. Efficacy Study: TNF-α-Induced Inflammation in HaCaT Cells

The potential of CBD-loaded nanoparticles stabilized with CA, LA, or SALA for treatment of skin disorders such as psoriasis was examined by their effect on TNF-α-induced HaCaT cells. We monitored the effect of these LSNs on secretions of two cytokines: IL-6 and IL-8. Sangiovanni et al. demonstrated that CBD and Cannabis Sativa extract (CSE, standardized to 5% CBD) inhibited TNF-α-induced NF-kB transcription in a dose-dependent manner in HaCaT cells, implying its potential efficacy in psoriasis treatment [3]. They found that CBD did not reduce the secretion of IL-8 from TNF-α-induced HaCaT cells when treated with CBD at concentrations below 5 µM. However, Petrosino et al. showed that CBD could reduce IL-8 secretion in a dose-dependent manner in HaCaT cells treated with poly-(I:C), an in-house-prepared inductor of atopic dermatitis [58]. In their work, there was a significant response at concentrations ranged between 5 µM and 20 µM. It seems, therefore, that there is a dose–response relationship between attenuation of IL-8 release and the applied dose of CBD when treated with poly-(I:C). To confirm the effectiveness of CBD on TNF-α-induced HaCaT cells above 5 µM, we have measured the attenuation of both psoriasis-related cytokines, IL-6, and IL-8, by free, non-encapsulated CBD. The results of this evaluation are provided in Figure 7A,B.

As shown in Figure 7A,B, CBD suppresses the release of both IL-6 and IL-8. At a concentration of 1 µM, CBD had almost no effect on IL-8 release compared to the positive control, while a clear pro-inflammatory effect was noted at this concentration as the IL-6 release was more than two times higher than the release by the positive control cells. However, with the increase in CBD concentration, the release of both cytokines decreased, reaching maximal effect at 4–8 µM CBD, and then the release of both cytokines increased again to a certain extent. This phenomenon may be related to the low solubility of CBD and its tendency to precipitate. The solubility of CBD in water is about 0.3 µM, and the solubility in the culture medium is 8–10 µM due to solubilization by BSA, which is present in a fully supplemented medium at 0.025% *w*/*w*. Thus, at concentrations above or close to 8–10 µM, CBD should exist in the treatment medium in both solubilized and precipitated forms, when the latter form elicits cytokine secretion from the cells. Several studies also reported this pro-inflammatory effect by evaluating the toxicity of insoluble nano-sized materials [59,60,61]. Particularly, Menas et al. [61] observed that the level of both IL-6 and IL-8 increased in response to treating the cells with cellulose nanocrystals.

Following the studies with free CBD, the effect of CBD-loaded LSNs on TNF-α-induced inflammation was evaluated in HaCaT cells. A comparison was made between the influence of CBD-loaded and CBD-unloaded LSNs on the release of cytokines with and without TNF-α induction. The unloaded LSNs served as an additional internal control for each type of LSN. This experimental design enabled us to distinguish between the possible inflammatory effect induced by the LSNs and the inflammation induced by TNF-α. Initial evaluations of CBD-loaded and unloaded LSNs showed only a marginal or insignificant effect on the release of cytokines, suggesting that availability of CBD in the treatment samples was below the effective concentration. The low CBD availability was due to its poor solubility in the cell culture medium. To increase CBD accessibility to the HaCaT cells, BSA was added into the culture medium as a solubilizer. Lehman et al. used 1.5% *w*/*w* BSA to improve vitamin D3 solubility in a HaCaT cell culture [62]. On the other hand, a concentration of 0.02% *w*/*w* (200 µg/mL) BSA was able to reduce the viability of HaCaT to about 80% [63]. It was therefore essential to determine the lowest level of BSA that sufficiently increases CBD availability during this cell assay. Dispersions of CBD-loaded CA-LSNs at a concentration of 40 µM CBD were tested for their load release in media containing various BSA concentrations. The experiments showed a CBD release of 8 µM (20% release) without BSA supplementation (0.025% *w*/*w* BSA), and 20 µM (50%), 23 µM (57.5%), 28 µM (70%), and 31 µM (77.5%), which were supplemented with 0.5%, 1.0%, 2.0% and 4.0% *w*/*w* BSA, respectively. Sufficient increase in solubility was obtained with 0.5% *w*/*w* BSA; therefore, this concentration was selected for further LSN evaluation. A cell culture medium with 0.5% BSA was used for all test groups and controls, except for the negative control that was not supplemented with BSA and was intended for evaluation of the influence of BSA on cytokine release.

The anti-inflammatory evaluation was performed at 20 × 10^9^ NPs/mL concentration, corresponding to the concentration of 50 μM CBD, which had been shown as non-toxic in Section 3.7.1. Figure 8A,B shows that all the LSN formulations had a similar effect on reducing the secretion of the two cytokines after induction of inflammation with TNF-α. Induction of HaCaT cells with TNF-α led to a 3–5 times increase in the release of both cytokines. Statistically significant reduction in the release of both IL-6 and IL-8 was observed after treatment with all CBD-loaded LSN formulations (*p* < 0.05 and *p* < 0.01, ANOVA test). Treatment with CBD-unloaded LSNs had not much effect if at all on the release of the cytokines and was similar to the release detected in the untreated controls of both the TNF-α-induced cells and the non-inflamed cells. Thus, the results have confirmed that the LSNs do not affect the release of these cytokines. Interestingly, in the cells that were not induced by TNF-α (Figure 8A.2), a higher release of IL-6 was observed in the control cells without BSA supplementation compared to the control cells supplemented with 0.5% BSA, which is in contrast with the results obtained for IL-8 secretion (Figure 8B.2).

IL-8 release in the control cells in a culture medium without BSA supplementation was half of the cytokine released in the control cells with BSA and was actually similar to the IL-8 release after treatment with CBD-loaded LSNs. That is, while IL-6 release is induced by BSA, IL-8 release was shown to be inhibited by BSA. The induced effect of BSA on the release of IL-6 may explain the relatively lower suppressing effect of CBD-loaded LSNs on the release of IL-6 compared to their effect on IL-8 release. These things considered, the extent of suppressing the release of both IL-6 and IL-8 was similar to the effect of free CBD at concentrations of 2 µM and 16 µM (Figure 7). In summary, CBD-loaded LSNs were effective when applied with 0.5% BSA in the cell culture medium. Bioassays using cells in a medium in which the solubility is limited for lipid-based formulations and hydrophobic active agents such as CBD require the use of solubilizing materials, otherwise, the results might be incorrect. In that context, it should be mentioned that several reviews have addressed the problematics of poor drug solubility that influences cell-involved assays [64,65,66].

## 3. Materials and Methods

### 3.1. Materials for Formulations and Chromatography

Cetyl Alcohol (CA), Triethyl citrate (TEC), Stearic Acid (SA), Lauric acid (LA), and Bovine Serum Albumin (BSA) were purchased from Sigma-Aldrich (Sigma-Aldrich Inc., St. Louis, MO, USA). Ethylcellulose (EC) grades std. 7 was purchased from Dow (Midland, MI, USA). Cannabidiol was obtained from Alfa Aesar (Heyshyam, UK). High-performance liquid chromatography (HPLC)–grade solvents were obtained from J.T. Baker (Mallinckrodt Baker, Inc., Phillipsburg, NJ, USA).

### 3.2. Materials for Cell Culture and LPS Treatment

An immortalized human keratinocytes line (HaCaT) was purchased from Cell Line Service GmbH (number 300493, Eppelheim, Germany). The cells were maintained in Dulbecco’s modified Eagle medium (DMEM, Gibco, Grand Island, NY, USA) supplemented with 10% fetal bovine serum (FBS, Gibco) at 37 °C under an atmosphere of 5% CO_2_ and 95% air. Human TNF-α was purchased from PeproTech (PeproTech Asia, Rehovot, Israel).

### 3.3. Nanoparticle Preparation

The stabilizing lipids, CA, SA, or SALA (SA/LA 24:76), each at a concentration of 0.025% *w*/*w*, were dissolved with 0.05% *w*/*w* EC, 0.05% *w*/*w* TEC, and 0.015% *w*/*w* CBD in 20 g of absolute ethanol. If LA alone was incorporated, then a 0.05% *w*/*w* concentration was used. The solution was kept at a constant stirring of 700 rpm. The ratio of the magnetic stirring bar’s length to the beaker diameter was 1:3. Deionized water was added by dripping at a constant rate of about 5.2 mL/min using a syringe pump (NE-300, New Era Pump Systems, Farmingdale, NY, USA) to the final content of about 60% *w*/*w* of the final dispersion mass. To form a pure aqueous suspension, ethanol in the obtained NP suspension was removed by evaporation with an R-205 Rotavapor (Büchi Labortechnik AG, Switzerland). To obtain a concentrated NP dispersion, several consecutive centrifugation steps were performed. For DSC analysis, the concentrated NPs dispersion was lyophilized with mannitol as a cryoprotective agent. The CBD content in the obtained nanoparticles was determined by HPLC. The overall process entrapment efficiency percentage (*EE*%) corresponding to the process yield was calculated according to the following equation:$$EE\%=\frac{Mass\;of\;CBD\;in\;the\;concentrated\;dispersion}{Total\;mass\;of\;CBD\;used}\times 100$$

### 3.4. Size and Microscopic Analysis

#### 3.4.1. Dynamic Light Scattering (DLS)

The hydrodynamic diameter spectrum of the NPs was collected using a CGS-3 Compact Goniometer System (ALV GmbH, Langen, Germany). The laser power was 20 mW at the HeNe laser line (632.8 nm). Correlograms were calculated by ALV/LSE 5003 correlator, which were collected at 90° angle, during 20 s for 10 repetitions, at 25 °C. The NP size was calculated using the Stokes–Einstein relationship, and the analysis was based on the regularization method as described by Provencher [67].

#### 3.4.2. Nanoparticle Tracking Analysis (NTA)

Measurements were performed using a NanoSight NS300 instrument (Malvern Instruments Ltd., Worcestershire, UK), equipped with a 632 nm laser module and 450 nm long-pass filter, and a camera operating at 25 frames per second, capturing a video file of the particles moving under Brownian motion. The software for capturing and analyzing the data (NTA 2.3) calculated the hydrodynamic diameters of the particles by using the Stokes–Einstein equation and their concentration (NPs/mL).

### 3.5. Differential Scanning Calorimetry (DSC)

Thermal analysis of NPs was performed using a Mettler-Toledo DSC1 STAR System (module DSC823e/700/484LN2, Mettler-Toledo AG, analytical, CH-8603 Scherzenbach, Switzerland). NP samples (2–3 mg) were weighed accurately in aluminum hermetic sample pans. The samples were analyzed between 30 °C and 200 °C at a ramp rate of 10 °C min^−1^.

### 3.6. Determination of CBD in NP Dispersion

CBD content within nano-sized particles was determined by adding 975 μL methanol to 25 μL aliquots from each carefully weighed NP sample, then stirred. After at least 10 min, the samples were further diluted 1:10 with methanol and injected into an HPLC system (1260 Infinity II, Agilent Technologies Inc., Santa Clara, CA, USA), equipped with a prepacked column (ReproSil-Pur 300 ODS-3, 5 µm, 250 mm 4.6 mm, Dr. Maisch, Ammerbuch, Germany), which was constantly maintained at 30 °C. The samples were chromatographed using a mobile phase consisting of acetonitrile-35 mM acetic acid (75:25) at a flow rate of 1 mL/min. Standard calibration curves, i.e., peak areas measured at 208 nm versus CBD concentrations, were constructed by running standard drug solutions in methanol for each series of chromatographed samples.

### 3.7. Cell Assays

#### 3.7.1. XTT Cell Viability and Proliferation Assay

HaCaT cells were seeded in 96 well plates at 2 × 10^4^ cells/well. After 24 h the cells were treated with CBD-loaded nanoparticles or unloaded nanoparticles stabilized with CA, LA or SALA dispersed in DMEM. After an additional 24 h, the XTT reagent (2,3-Bis-(2-Methoxy-4-Nitro-5-Sulfophenyl)-2*H*-Tetrazolium-5-Carboxanilide) was added, and after 1–2 h development, the plate was scanned by a Microplate reader (iMark, Bio-Rad) at 490 and 655 nm.

#### 3.7.2. HaCaT Cells Inflammation Induction with TNF-α

HaCaT cells were seeded in 24 well plates at 10^5^ cells/well. After 24 h the cells were treated for 2 h with 10 ng/mL TNF-α to induce inflammation. Afterwards, the medium was removed, the cells were treated with CA-, LA-, or SALA-stabilized NPs, either CBD-unloaded or CBD-loaded at concentrations of 20 × 10^9^ NPs/mL. The NPs were dispersed in DMEM with or without 0.5% BSA *w*/*w*. After an additional 24 h, the supernatants were collected for further analysis of IL-6 and IL-8 by an enzyme-linked immunosorbent assay (ELISA). The ELISA was performed according to the manufacturer’s instructions of the kit (ELISA Max, Biolegend, San Diego, CA, USA). The obtained IL-6 and IL-8 values were further normalized by protein content.

The HaCaT cells were carefully detached from the plate with trypsin, collected, and lysed on ice in deionized water with the aid of an Ultrasonic processor (Sonics, Vibra-cell VCX130, Newtown, CT, USA) at 50% amplitude with two 10 s pulses. The lysate from each well was analyzed with Bradford assay using Protein Assay Dye Reagent Concentrate (Bio-Rad laboratories, Hercules, CA, USA) for total protein estimation.

### 3.8. In Vitro Release Study

The in vitro release study was performed in 125 mL of release medium constituting PBS with 0.05% Tween 80 or DMEM maintained at 37 °C with magnetic stirring (500 RPM). Appropriate amounts of NPs were added at t = 0 to obtain CBD concentration of 1 µg/mL. At predetermined intervals, samples were withdrawn, immediately filtered through a 0.22 µ filter (Durapor, Merck-Millipore, Burlington, MA, USA), and further analyzed by HPLC (see Section 3.6). Removal of NPs by filtration was validated by comparative DLS analysis of the tested media before and after filtration. The analysis showed a reduction (by five orders of magnitude) in the intensity of the particle size of the filtered samples, indicating an almost complete removal of the NPs.

### 3.9. In Vitro Skin Penetration Study

The penetration of CBD from CBD-loaded NP into the skin was determined in vitro using a Franz diffusion cell system (Permegear, Inc., Bethlehem, PA, USA). The diffusion area was 1.767 cm^2^ (15 mm diameter orifice), and the receptor compartment volume was 12 mL. The solutions in the water-jacketed cells were thermostated at 37 °C and stirred by externally driven, Teflon-coated magnetic bars [68,69]. Each set of experiments was performed with twelve diffusion cells containing abdominal rat skin. Sprague-Dawley rats (males, 200–300 g) were euthanized by aspiration of CO_2_. The animal treatments were performed in accordance with protocol reviewed and approved by the Institutional Committee for the Ethical Care and Use of Animals in Experiments, Ben-Gurion University of the Negev, which complies with the Israeli Law of Human Care and Use of Laboratory Animals, authorization number: IL-30-06-2020 (C). The abdominal hair was carefully clipped, and sections of full-thickness skin were excised from the fresh carcasses of animals and used immediately. All skin sections were measured for transepidermal water loss (TEWL), and only those pieces with TEWL measurements less than 10 g/m^2^/h were used. TEWL testing was performed on skin pieces using a Dermalab Cortex Technology instrument, (Hadsund, Denemark). The skin was placed on the receiver chambers with the stratum corneum facing upwards, and the donor chambers were then clamped in place. The receiver chamber, defined as the side facing the dermis, was filled with phosphate buffer (pH 7.4)—ethanol 50:50 solution to allow a sink condition [70]. Aqueous dispersions (0.2 mL) of lipid-stabilized NPs, each containing 100 μg of entrapped CBD, were applied on the skin at time = 0. At t = 2 h, 4 h, and 6 h, 1 mL samples were withdrawn from the receiver and analyzed by HPLC (see Section 3.6). The receiver solutions were carefully replenished with fresh buffer—ethanol 50:50 solution. After the 6-h experimental period, each exposed skin tissue was removed, washed with plenty of water, wiped carefully, and tape-stripped (×15) to remove CBD adsorbed in the stratum corneum. Penetrated levels in the skin tissues were determined after overnight methanol extraction by HPLC (see Section 3.6).

### 3.10. Hansen Solubility Parameter (HSP)

The solubility model parameters are: $\delta_{D}$ denoted as the “dispersion” or Van der Waals component of a molecule, $\delta_{P}$ is the “polar” component, and $\delta_{H}$ is the “hydrogen-bonding” component. HSP provides an objective “distance” measure for likeness or similarity between two molecules in 3D space. Given two chemicals with HSP [$\delta_{D1}$, $\delta_{P1}$, $\delta_{H1}$] and [$\delta_{D2}$, $\delta_{P2}$, $\delta_{H2}$], the HSP distance is given by: $R_{a}=\sqrt{4{\left(\delta_{D1}-\delta_{D2}\right)}^{2}+{\left(\delta_{P1}-\delta_{P2}\right)}^{2}+{\left(\delta_{H1}-\delta_{H2}\right)}^{2}}$. The HSP for a mixture of substances can be calculated for each of the components, *D*, *P*, and *H*, as $\delta_{mix}=\underset{i}{\sum }\delta_{i}\varphi_{i},$ where $\varphi_{i}$ is the fraction of a substance in the mixture.

## 4. Conclusions

CBD-loaded EC-based LSNs (220–260 nm in diameter, 70–85% entrapment efficiency) were characterized and shown to lack any crystalline regions, implying a uniform distribution of molecular structures in the nanoparticles. In addition, it was possible to show a relationship between the thermal characterization and CBD release profile, showing an effect of the stabilizing lipid. Since the SALA melting point (36 °C) was below the experimental temperature compared to CA (51 °C), it is assumed that SALA was released into the system in its liquid form, facilitating the solubilization of CBD. We have shown that EC-based LSNs were able to deliver CBD into and through rat skin in an in vitro diffusion cell model. The LSNs also showed selective permeation and dissolution properties with respect to the incorporated stabilizing lipid. The melting point of these excipients significantly influenced the permeation and its further physicochemical stability of CBD at the application site. The CA-, LA-, and SALA-stabilized NPs showed a better skin retention of CBD (>3.0 μg/cm^2^) compared to nanoparticles stabilized with SA (2 μg/cm^2^), a lipid with a higher melting point relative to the others. Since CA and its formulation have the lowest affinity to CBD, SA-stabilized NPs loaded with CBD have the highest activity in terms of CBD saturation, and consequently, a significantly high percutaneous permeability was obtained after 6 h in a skin permeation study relative to the other LSNs. Considering the applicability of the CBD-loaded LSNs for the treatment of inflammatory conditions, and particularly for psoriasis, it was shown that CBD containing LSNs reduced TNF-α-induced release of pro-inflammatory cytokines, IL-6 and IL-8, in a HaCaT cell model when applied in combination with BSA as a release mediator. It was shown that both cytokines were suppressed by 20–40% after treatment with CBD-loaded LSNs relative to the untreated inflamed cells. These findings emphasize the vital role of stabilizing lipids in CBD-containing nanoparticulate systems. Further in vivo studies remain to be done to practically evaluate LSN potential in pre-clinical and clinical trials.

## Figures and Tables

**Figure 1 molecules-28-01774-f001:**
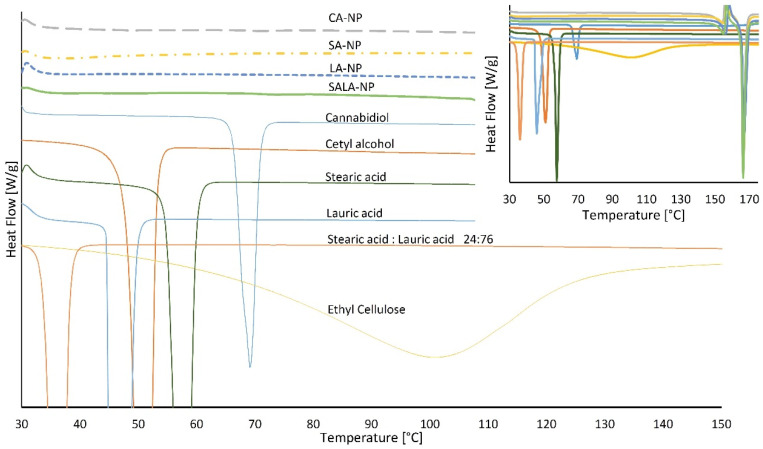
Differential scanning calorimetry profiles of lipid-stabilized nanoparticles and their sourced ingredients. The upper right corner presents the full-range thermogram with mannitol melting peaks at 169 °C.

**Figure 2 molecules-28-01774-f002:**
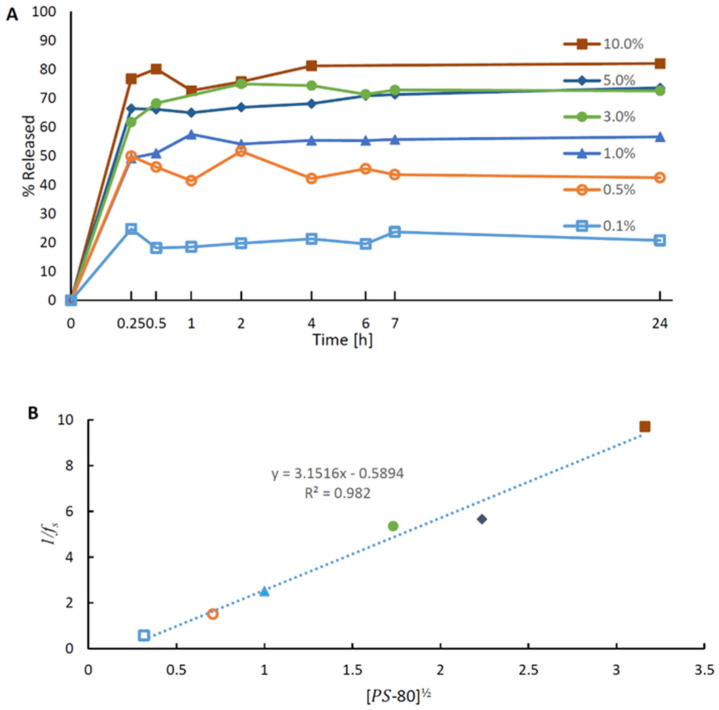
CBD release profiles from cetyl alcohol-stabilized nanoparticles: (**A**) release at various concentrations of polysorbate 80 in PBS solution; and (**B**) relationship between the reciprocal of the fraction of the precipitated CBD after 24 h and the square root (a=2) of polysorbate concentrations (according to Equation (5)).

**Figure 3 molecules-28-01774-f003:**
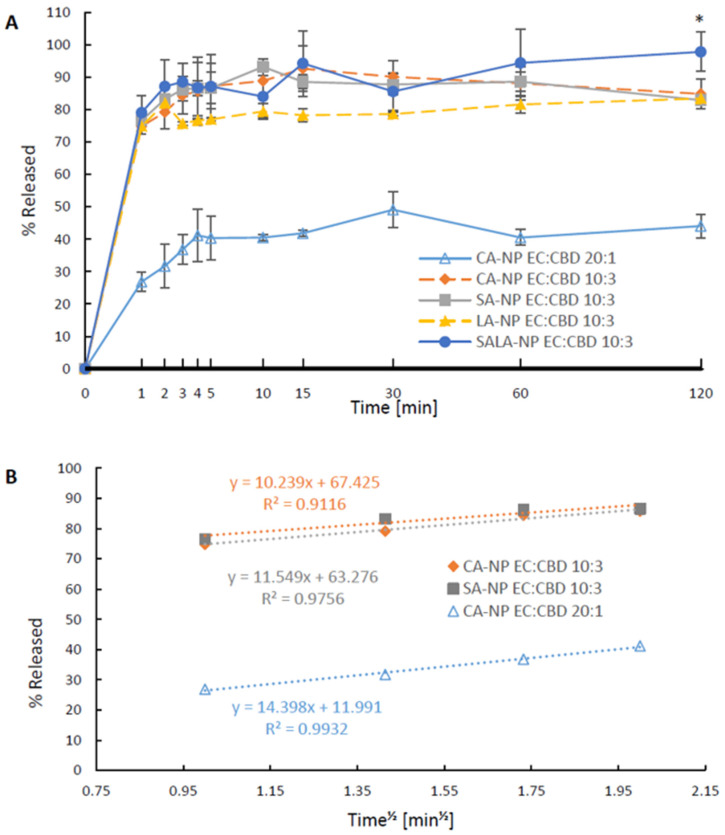
CBD release from LSNs: (**A**) comparison between CA-NPs, SA-NPs, LA-NPs, SALA-NPs, and CA-NPs containing a relatively higher ratio of ethyl cellulose/CBD, * *p* value < 0.05 between SALA-NP and other groups. *p* values calculated by one-way ANOVA, followed by post hoc Tuckey test at 120 min interval, values are presented as mean ± SD; and (**B**) Linear CBD release patterns from LSNs fitted the Higuchi model.

**Figure 4 molecules-28-01774-f004:**
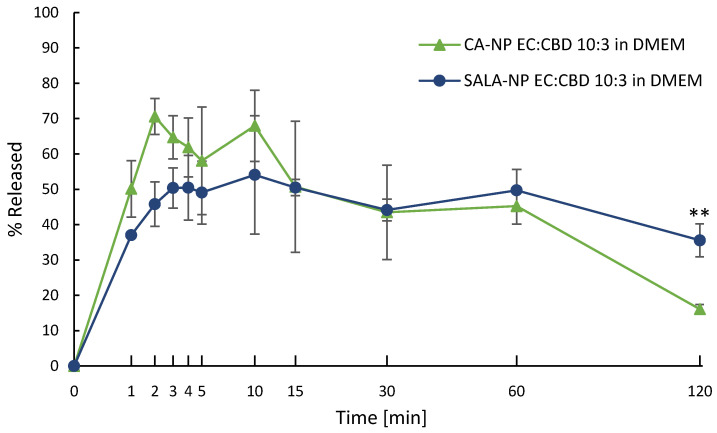
CBD release from LSNs in DMEM cell culture medium, ** *p* value < 0.01 between CA-NP and SALA-NP (calculated as a 2-tailed homoscedastic Student’s *t*-test) at 120 min interval. Values are presented as mean ± SD.

**Figure 5 molecules-28-01774-f005:**
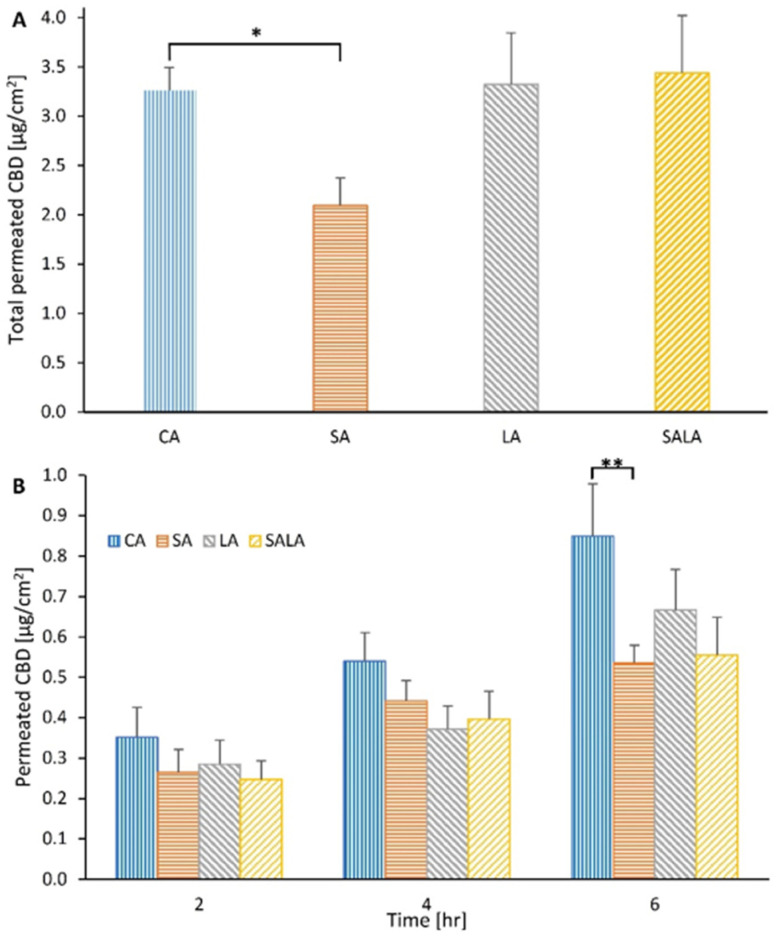
Skin retention of CBD after 6-h application: (**A**) * *p* value < 0.05 between compared groups, and percutaneous permeation of CBD delivered via lipid-stabilized nanoparticles; and (**B**) ** *p* value < 0.01 between compared groups. The *p*-value was calculated as a two-tailed homoscedastic Student’s *t*-test. Values are presented as mean ± SEM.

**Figure 6 molecules-28-01774-f006:**
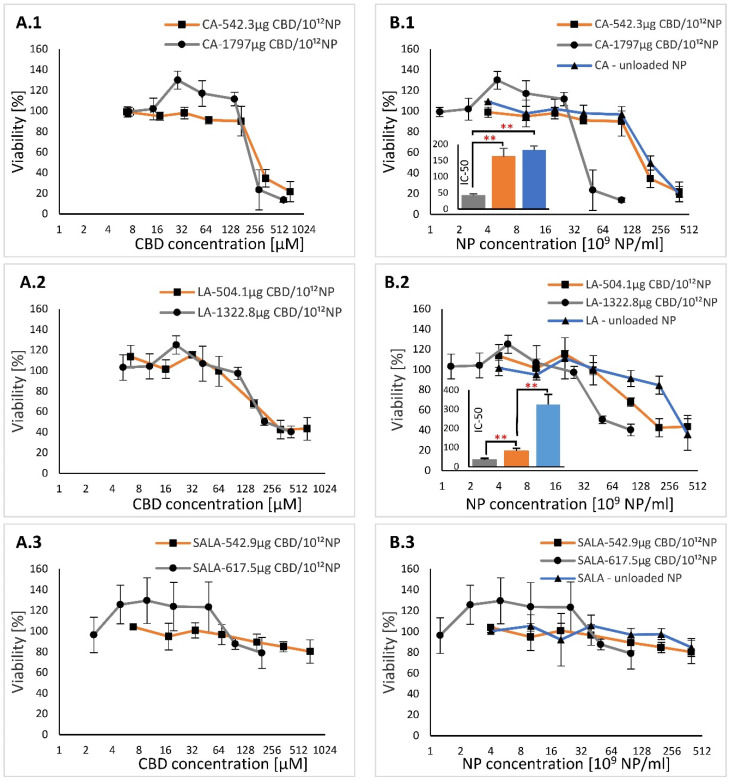
Viability profiles of HaCaT cells after incubation with various types of LSNs: (**A.1–A.3**) % viability plotted against CBD concentrations; and (**B.1–B.3**) % viability plotted against NP concentrations. The inserts in B.1 and B.2 show IC-50 values with respect to NP concentration. ** *p* value < 0.01. The *p*-values were calculated by one-way ANOVA, followed by a post hoc Tuckey test. Values are presented as mean ± SD.

**Figure 7 molecules-28-01774-f007:**
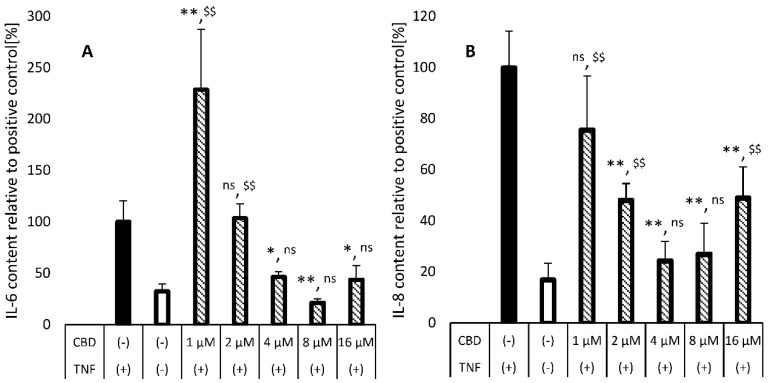
IL-6 (**A**) and IL-8 (**B**) secretion from TNF-α-induced inflamed HaCaT cells treated with various concentrations of CBD. The first two columns (filled and blank bars) represent CBD-free control groups with and without TNF-α induction. * indicates comparison to the positive control TNF(+). ^$$^ indicates comparison to the negative control TNF(−). * *p* value < 0.05, **/^$$^
*p* value < 0.01, ns—no significant difference between compared groups. The *p* values were calculated by one-way ANOVA, followed by the Tuckey test. Values are presented as mean ± SD.

**Figure 8 molecules-28-01774-f008:**
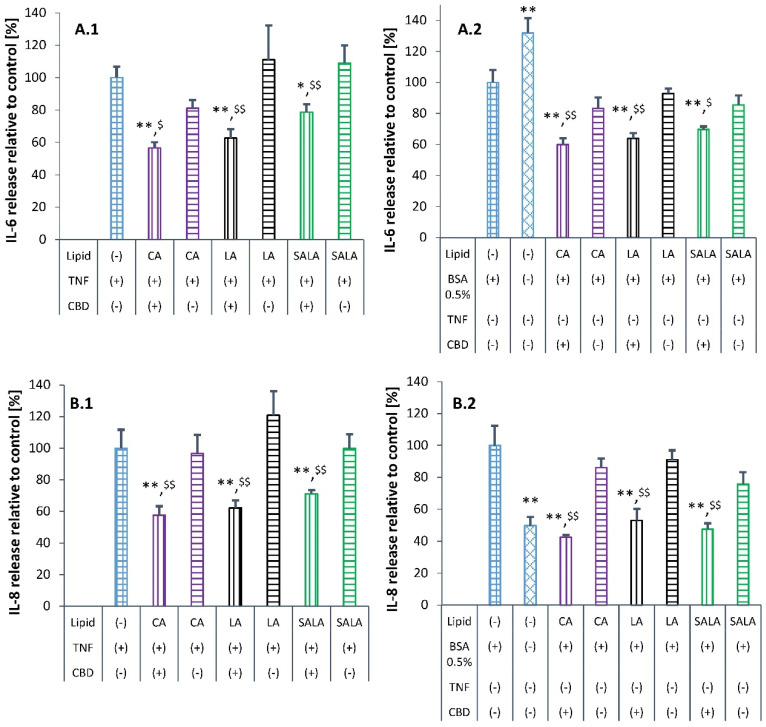
Influence of lipid-stabilized nanoparticles on IL-6 (**A.1**,**A.2**) and IL-8 (**B.1**,**B.2**) secretion in TNF-α-induced and non-induced HaCaT cells. * Indicates comparison to control (100%). ^$^ indicates comparison to unloaded LSNs of the same type. */^$^
*p* value < 0.05, **/^$$^
*p* value < 0.01. The *p* values calculated by one-way ANOVA, followed by post hoc Tuckey test. Values are presented as mean ± SD.

**Table 1 molecules-28-01774-t001:** Nanoparticle size and entrapment efficiency EE%.

	Diameter [nm]	PDI	EE%
NP-CA	223 ± 13	0.018 ± 0.014	85.7 ± 15.6
NP-CA-u	225 ± 14	0.074 ± 0.060	---
NP-SA	266 ± 8	0.102 ± 0.078	73.7 ± 21.4
NP-SA-u	262 ± 17	0.025 ± 0.027	---
NP-LA	244 ± 12	0.044 ± 0.021	79.3 ± 18.1
NP-LA-u	246 ± 7	0.094 ± 0.062	---
NP-SALA	235 ± 27	0.106 ± 0.069	84.4 ± 17.5
NP-SALA-u	235 ± 11	0.067 ± 0.033	---

CA = cetyl alcohol, SA = stearic acid, and LA = lauric acid. “u” means unloaded NPs; PDI = polydispersity index.

**Table 2 molecules-28-01774-t002:** Thermal transition parameters of nanoparticle components.

Component	Thermal Transition Observed [°C]	Transition [°C] Acc. to the References	Transition Type	Reference
**Cannabidiol**	69.10	66–67, 71	m.p.	[36,37]
**Ethyl cellulose**	100.47	129–133	Tg	[38]
**Cetyl alcohol**	50.96	50.81	m.p.	[39]
**Stearic acid**	57.60	54/69.50	B to C form transition/m.p.	[40]
**Lauric acid**	43.60	46.36	m.p.	[30]
**Stearic-Lauric acid 24:76**	36.00	38.99	m.p.	[30]

**Table 3 molecules-28-01774-t003:** Hansen solubility parameters (HSP) and the δ values of human skin, LSNs, and their ingredients.

Component	Mixture Ratio	$\delta_{D}$	$\delta_{P}$	$\delta_{H}$	R_a_-CBD	R_a_-Skin
Human skin [56]	---	17	8	8	5.33	---
Cannabidiol ^1^ [55]	---	18.3	4.8	11.3	---	5.33
Cetyl alcohol [55]	---	15.1	3.7	8.1	7.3	5.74
Stearic acid [57]	---	16.3	3.3	5.5	7.2	5.50
Lauric acid [57]	---	16.2	4.0	7.4	5.8	4.35
Stearic-Lauric acids 24:76 ^2^	24:76	16.2	3.8	6.9	6.1	4.57
**NP Formulations**						
CBD-CA ^2^	5:3	16.30	4.10	9.31	---	4.34
CBD-SA ^2^	5:3	17.05	3.85	7.68	---	4.16
CBD-LA ^2^	10:3	16.69	4.18	8.30	---	3.89
CBD-SALA ^2^	5:3	17.01	4.18	8.58	---	3.86

^1^—calculated according to functional groups contribution method; ^2^—calculated according to materials mixture formula.

## Data Availability

Data are available upon request.

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
