# Peer review of "The Fundamental Role of Lipids in Polymeric Nanoparticles: Dermal Delivery and Anti-Inflammatory Activity of Cannabidiol"

_molecules, 2023, doi:10.3390/molecules28041774_

Round 1

Reviewer 1 Report

Reviewer’s Comments:

The manuscript “The Fundamental Role of Lipids in Polymeric Nanoparticles: Dermal Delivery and Anti-Inflammatory Activity of Cannabidiol” is very interesting work. This report presents a nanoparticulate platform for cannabidiol (CBD) with a potential to treat topical inflammatory conditions. We have previously shown that stabilizing lipids improve the encapsulation of CBD in ethyl cellulose nanoparticles. In this study, we examined CBD release, skin permeation, and the capability of lipid-stabilized nanoparticles (LSNs) to suppress the release of IL-6 and IL-8. The nanoparticles were stabilized with cetyl alcohol (CA), stearic acid (SA), lauric acid (LA), and SA/LA eutectic combination (SALA). Following measurements of particle size, polydispersity index, LSN concentrations, differential scanning calorimetry (DSC) characterization, and HPLC analysis, the in vitro release of CBD was performed in phosphate buffered saline (PBS) and cell culture medium. Skin permeability was performed in Franz diffusion cells mounted with fresh rat skin. However, the following issues should be carefully treated before publication.

1. In abstract, the author should add more scientific findings.

2. Keywords: the synthesized system is missing in the keywords. So, modify the keywords.

3. In the introduction part, the introduction part is not well organized and cited references should cite recently published articles such as 10.3390/molecules27196457, 10.3390/molecules27196564

4. Introduction part is not impressive and systematic. In the introduction part, the authors should elaborate the scientific issues in the Cannabidiol research.

5. Nanoparticle Preparation…, The author should provide reason about this statement “Ethanol in the obtained NP suspension was evaporated by R-205 Rotavapor (Büchi Labor-technik AG, Switzerland)”.

6. The authors should explain regarding the recent literature why “The solutions in the water-jacketed cells were thermostated at 37°C and stirred by externally driven, Teflon-coated magnetic bars”.

7. Nanoparticle preparation method and characterization. The author should explain the latest literature “As seen in Table 1, not much difference in size was obtained between the loaded and unloaded NPs stabilized with the same lipid.”.

8. The author should provide reason about this statement, “Unlike a gel or an ointment, which can be massaged on the skin to increase penetration, the unstirred volume of these nanodispersions (0.2 mL) was relatively excessive for normal dermal application as only a small portion of LSNs came in direct contact with the skin”.

9. Comparison of the present results with other similar findings in the literature should be discussed in more detail. This is necessary in order to place this work together with other work in the field and to give more credibility to the present results.

10. The conclusion part is very week. Improve by adding the results of your studies.

Author Response

To the Assistant Editor

Molecules

Dear Ms. Ewa Myszka,

We want to thank you and the reviewers for the useful comments. We modified our paper accordingly. With this letter, I also include the revised version of the manuscript entitled “The Fundamental Role of Lipids in Polymeric Nanoparticles: Dermal Delivery and Anti-Inflammatory Activity of Cannabidiol” by Zamansky et al.  (Ms. Ref. No. molecules-2174445).

Response to Reviewer 1:

  1. In the abstract, the author should add more scientific findings. The abstract has been changed accordingly (in the required frame of about 200 words).
  2. Keywords: the synthesized system is missing in the keywords. So, modify the keywords. The synthesized system is defined as lipid-stabilized nanoparticles (LSNs) and it is included in the Keywords.
  3. In the introduction part, the introduction part is not well organized and cited references should cite recently published articles such as 10.3390/molecules27196457, 10.3390/molecules27196564. We want to thank the reviewer for the recent publications. We, of course, cited and included them in the reference list.
  4. Introduction part is not impressive and systematic. In the introduction part, the authors should elaborate the scientific issues in the Cannabidiol research. The introduction part has been changed and organized according to the reviewer’s comments. The previous research on Cannabidiol was input and elaborated at the beginning of this part.
  5. Nanoparticle Preparation…, The author should provide reason about this statement “Ethanol in the obtained NP suspension was evaporated by R-205 Rotavapor (Büchi Labor-technik AG, Switzerland)”. The sentence was changed on p. 5: “To form a pure aqueous suspension, ethanol in the obtained NP suspension was removed by evaporation with R-205 Rotavapor (Büchi Labortechnik AG, Switzerland)”.
  6. The authors should explain regarding the recent literature why “The solutions in the water-jacketed cells were thermostated at 37°C and stirred by externally driven, Teflon-coated magnetic bars”. Two recent references were cited regarding the techniques used with the Franz diffusion cell system (p. 8).
  7. Nanoparticle preparation method and characterization. The author should explain the latest literature “As seen in Table 1, not much difference in size was obtained between the loaded and unloaded NPs stabilized with the same lipid.”. An explanation was added in p. 9, 2nd paragraph: “As seen in Table I, no profound differences in size were obtained between the CBD-loaded and unloaded NPs stabilized with the same lipid, which indicate that the creation of the nanoparticle network is not affected by small molecules. However, it seems that the nature of the stabilizing lipid directly influences the obtained NP size, particularly less water-soluble fatty alcohol CA and more soluble fatty acids LA, SA and their combination SALA. In our previous work, we have shown that inclusion of materials with lower water solubility contributes to NP size reduction by creating larger number of nucleation centers [26]. Thus, the polymer is distributed between large number of particles at a smaller size”.
  8. The author should provide reason about this statement, “Unlike a gel or an ointment, which can be massaged on the skin to increase penetration, the unstirred volume of these nanodispersions (0.2 mL) was relatively excessive for normal dermal application as only a small portion of LSNs came in direct contact with the skin”. The reason is that the current model provided by the diffusion cell systems is unable to mimic the real application of dermal products. We pointed that out on p. 17, section 3.3: “Usually, topical formulations are applied at 4-25 mg/cm2, thus unlike a gel or an ointment, which can be massaged on the skin to increase penetration, the unstirred volume of these nano-dispersions (0.2 mL) was relatively excessive for normal dermal application as only a small portion of LSNs came in direct contact with the skin [55].
  9. Comparison of the present results with other similar findings in the literature should be discussed in more detail. This is necessary in order to place this work together with other work in the field and to give more credibility to the present results. This paper describes a nano-system based on ethyl cellulose-lipid hybrid for topical/dermal delivery of cannabidiol. Except our previous paper (ref. 26) and Hassan et al. (2023, “Formulation and development of novel lipid-based combinatorial advanced nanoformulation for effective treatment of non-melanoma skin cancer” doi: 10.1016/j.ijpharm.2022.122580), no other reports have been published so far on the dermal application of cannabidiol in nanoparticles (PubMed keywords: nanoparticles-cannabidiol-dermal-skin). However, we included as many references as we found dealing with the anti-inflammatory activity of cannabidiol, supporting its potential for treating dermatological disorders when applied in nanoparticulate systems.
  10. The conclusion part is very weak. Improve by adding the results of your studies. Thanks for this comment. The conclusion section was improved and expanded as recommended by the reviewer.

With best regards,

Prof. Shimon Ben-Shabat

Prof. Amnon Sintov

Reviewer 2 Report

Dear Authors,

The work that you present in your manuscript is very interesting, regarding the use of cannabidiol in nanoparticles, the dermal delivery and the anti-inflammatory potential.

The information is very well presented, the discussions are well described and detailed, the figures easy to understand.

Still, I suggest you organise the large paragraphs in each section into smaller paragraphs, in order to be easily read and understand. 

Plus, I suggest that you use an anti-inflammatory drug used in therapy, especially in dermal therapy.

Author Response

To the Assistant Editor

Molecules

Dear Ms. Ewa Myszka,

We want to thank you and the reviewers for the useful comments. We modified our paper accordingly. With this letter, I also include the revised version of the manuscript entitled “The Fundamental Role of Lipids in Polymeric Nanoparticles: Dermal Delivery and Anti-Inflammatory Activity of Cannabidiol” by Zamansky et al.  (Ms. Ref. No. molecules-2174445).

Response to Reviewer 2:

The work that you present in your manuscript is very interesting, regarding the use of cannabidiol in nanoparticles, the dermal delivery and the anti-inflammatory potential.

The information is very well presented, the discussions are well described and detailed, the figures easy to understand. Thanks!!

Still, I suggest you organise the large paragraphs in each section into smaller paragraphs, in order to be easily read and understand. Each section was separated into smaller paragraphs.  

Plus, I suggest that you use an anti-inflammatory drug used in therapy, especially in dermal therapy. Indeed, we will include in our in vivo experiments a group of animals (psoriasis-like model) applied with an anti-inflammatory drug, e.g., corticosteroids.

With best regards,

Prof. Shimon Ben-Shabat

Prof. Amnon Sintov

Reviewer 3 Report

This report discusses a new method for delivering CBD to the skin to alleviate inflammation. The CBD is incorporated into nanoparticles which makes it easier for the skin to absorb it. The report examines the use of different types of compounds to stabilize the nanoparticles, and how this affects the amount of CBD that is released and how well it is able to penetrate the skin. It also studies the impact of these nanoparticles on the release of two chemicals, IL-6 and IL-8, which are associated with inflammation. The study found that the nanoparticles effectively reduced the release of these chemicals, suggesting that CBD-loaded nanoparticles may be useful in treating topical inflammatory conditions.

The study needs to be rewritten and restructured. It lacks a clear focus on the adjustment of impact of CBD on release. The results suggest that the impact of CBD on release is influenced by crystal formation and the addition of different chemicals, but the study does not provide a comprehensive approach to addressing the complexities of release. This is a complex subject and experimentation alone is not sufficient to account for factors such as crystal formation, temperature, matrix conditions, dispersion, and component distribution. The manuscript requires a comprehensive and cohesive approach that addresses these complexities in order to be coherent and effective.

Author Response

To the Assistant Editor

Molecules

Dear Ms. Ewa Myszka,

We want to thank you and the reviewers for the useful comments. We modified our paper accordingly. With this letter, I also include the revised version of the manuscript entitled “The Fundamental Role of Lipids in Polymeric Nanoparticles: Dermal Delivery and Anti-Inflammatory Activity of Cannabidiol” by Zamansky et al.  (Ms. Ref. No. molecules-2174445).

Response to Reviewer 3:

This report discusses a new method for delivering CBD to the skin to alleviate inflammation. The CBD is incorporated into nanoparticles which makes it easier for the skin to absorb it. The report examines the use of different types of compounds to stabilize the nanoparticles, and how this affects the amount of CBD that is released and how well it is able to penetrate the skin. It also studies the impact of these nanoparticles on the release of two chemicals, IL-6 and IL-8, which are associated with inflammation. The study found that the nanoparticles effectively reduced the release of these chemicals, suggesting that CBD-loaded nanoparticles may be useful in treating topical inflammatory conditions.

 The study needs to be rewritten and restructured. It lacks a clear focus on the adjustment of impact of CBD on release. The results suggest that the impact of CBD on release is influenced by crystal formation and the addition of different chemicals, but the study does not provide a comprehensive approach to addressing the complexities of release. This is a complex subject and experimentation alone is not sufficient to account for factors such as crystal formation, temperature, matrix conditions, dispersion, and component distribution. The manuscript requires a comprehensive and cohesive approach that addresses these complexities in order to be coherent and effective. We disagree with this comment. The only “complexity” of the release is that the in vitro model is at a non-sink condition, due to the obvious precipitation of CBD after its release from the nanoparticles (not necessarily in a crystal form). Like many known techniques published previously, we encountered this obstacle by protein binding in the medium using BSA that adsorbs CBD. Incorporating protein in the medium imitates physiological fluids while not affecting the nanoparticles.

With best regards,

Prof. Shimon Ben-Shabat

Prof. Amnon Sintov

Round 2

Reviewer 3 Report

Although the document is well-aimed, it still has to be rebuilt.

Author Response

Response to the academic editor’s comments:

  1. Figures 2A+2B (and Figures 3A+B) have been corrected so no overlapping should occur.
  2. Thanks, the equation was changed according to the editor’s comment and appropriate citations were added:

Considering that  , where  is the concentration of polysorbate 80,  is the CMC value of polysorbate 80,  is its aggregation number (experimentally reported values ranging between 22-350  [51]), and  (  value is about 0.0014% w/w [52])  equations (3) and (4) can be combined:

  1. Figure 3B: time (min½) in the x-axis was corrected to time½ (min½). Thank you for noticing.

The validity of the model is really limited to cumulative release of up to 85% (maximum values in figure 3B are about 86.5%. We also added a short explanation on p. 15 (second paragraph): “The release pattern which was fitted to the Higuchi model implies that the NPs are formed as a polymeric matrix containing an evenly distributed active agent. This contrasts with a CBD core entrapped by a polymeric shell, in which the release should be linear as long as Cinitial>>Cs. The slower release as a function of polymer content has also been demonstrated in several publications describing release from EC microparticles and nanoparticles [54–56]. It can be seen that CBD release from the NPs with the higher content of polymer (EC:CBD 20:1, blue triangles) fits relatively better to the Higuchi model, probably due to their larger size and more likely due to the uniform distribution of CBD as also demonstrated by the DSC measurements (Figure 1)”.  

  1. Indeed, not many reports have been published so far on dermal application of cannabidiol in nanoparticles. When the keywords “nanoparticles-cannabidiol-dermal-skin” are searched by PubMed only 1 reference appears. Anyway, we added as many references as we found that fitted to the scope of this article.
